# Evolutionary games on isothermal graphs

Benjamin Allen [1,6]*, Gabor Lippner[2,6] & Martin A. Nowak[3,4,5]

Population structure affects the outcome of natural selection. These effects can be modeled using evolutionary games on graphs. Recently, conditions were derived for a trait to be favored under weak selection, on any weighted graph, in terms of coalescence times of random walks. Here we consider isothermal graphs, which have the same total edge weight at each node. The conditions for success on isothermal graphs take a simple form, in which the effects of graph structure are captured in the 'effective degree'—a measure of the effective number of neighbors per individual. For two update rules (death-Birth and birth-Death), cooperative behavior is favored on a large isothermal graph if the benefit-to-cost ratio exceeds the effective degree. For two other update rules (Birth-death and Death-birth), cooperation is never favored. We relate the effective degree of a graph to its spectral gap, thereby linking evolutionary dynamics to the theory of expander graphs. Surprisingly, we find graphs of infinite average degree that nonetheless provide strong support for cooperation.

[1] Department of Mathematics, Emmanuel College, 400 The Fenway, Boston, MA 02115, USA. [2] Department of Mathematics, Northeastern University, 360 Huntington Ave, Boston, MA 02115, USA. [3] Program for Evolutionary Dynamics, Harvard University, One Brattle Square, Cambridge, MA 02138, USA. [4] Department of Mathematics, Harvard University, One Oxford St, Cambridge, MA 02138, USA. [5] Department of Organismic and Evolutionary Biology, Harvard University, 26 Oxford St, Cambridge, MA 02138, USA. [6]These authors contributed equally: Benjamin Allen, Gabor Lippner
*email: allenb@emmanuel.edu

The structure of a population has important consequences for its evolution[1–10]. In particular, spatial or social network structure can promote the evolution of cooperative behavior, by allowing cooperators to cluster together and share benefits[11–13].

Spatial structure can be represented mathematically as a graph or network, in which nodes represent individuals and edges indicate spatial or social connections[6,14–20]. Edges can be weighted to indicate the strength of the connection. To study cooperation or other forms of social behavior, interactions can be modeled as matrix games. Individuals play games with their neighbors, and the payoffs from these games determine reproductive success.

Mathematical studies of evolutionary games on graphs[16–18,20–29] have typically assumed that the graph is regular, meaning that each individual has the same number of neighbors. Recently, a condition was derived that determines which strategy is favored in any two-player, two-strategy game, on any weighted graph, under weak selection[30–32]. Weak selection means that the game has only a small effect on reproductive success. For nonweak selection, determining the outcome of evolutionary games on graphs is PSPACE-complete[33].

A weighted graph is called isothermal if the sum of edge weights is the same at each vertex (Fig. 1). This property has a natural interpretation: suppose that the edge weights represent the amount of time that two individuals interact with each other. Then the graph is isothermal as long as each individual devotes the same total time to interaction. Importantly, some individuals may divide their time thinly among many contacts, while others focus primarily on one or two contacts.

Isothermal graphs have special relevance for evolutionary dynamics. All vertices of an isothermal graph have the same reproductive value—meaning that each vertex contributes equally to the future population under neutral drift[10,34]. The Isothermal Theorem[6,35] states that isothermal graphs neither amplify nor suppress the effects of selection for mutations of constant fitness effect.

Here, we analyze evolutionary games on isothermal graphs, and obtain more powerful results than are available for arbitrary weighted graphs[30–32]. The condition for a strategy to be favored, under weak selection, takes a particularly simple form, in which the graph structure is summarized in a single statistic, which we term the 'effective degree', $\tilde{\kappa}$. An isothermal graph of effective degree $\tilde{\kappa}$ behaves like an unweighted $\tilde{\kappa}$-regular graph in its effect on strategy selection. In particular, cooperation is favored on a large graph (for particular update rules; see below) if and only if the benefit to others exceeds $\tilde{\kappa}$ times the cost. We derive bounds on $\tilde{\kappa}$ in terms of the graph's spectral gap (the difference between the two largest eigenvalues), establishing a link to the theory of expander graphs[36–38]. Applying our results to power-law networks and to heterogeneous subdivided populations, we exhibit graphs of arbitrarily large average degree that provide arbitrarily strong support to cooperation.

## Results

**Model.** We represent spatial structure by a weighted, connected, isothermal graph $G$ of size $N$. The edge weight between vertices $i, j \in G$ is denoted $w_{ij}$. Without loss of generality, we scale edge weights so that $\sum_{j \in G} w_{ij} = 1$ for each vertex $i$. In this way, edge weights may be interpreted as probabilities or frequencies of interaction. Edges are undirected, meaning $w_{ij} = w_{ji}$, and there are no self-loops: $w_{ii} = 0$ for each $i$. Two vertices are neighbors if they are joined by an edge of positive weight; the number of neighbors of vertex $i$ is called its topological degree, $k_i$.

Vertices in an isothermal graph may differ widely in the distribution of edge weights among their neighbors (Fig. 1). We quantify these differences using the 'Simpson degree'[20] (Fig. 2), defined for each vertex $i$ as

$$\kappa_i = \left( \sum_{j \in G} w_{ij}^2 \right)^{-1}. \tag{1}$$

In words, if individual $i$ randomly selects two neighbors, with probability proportional to edge weight, then $\kappa_i$ is the inverse probability that the same neighbor is selected twice. The Simpson

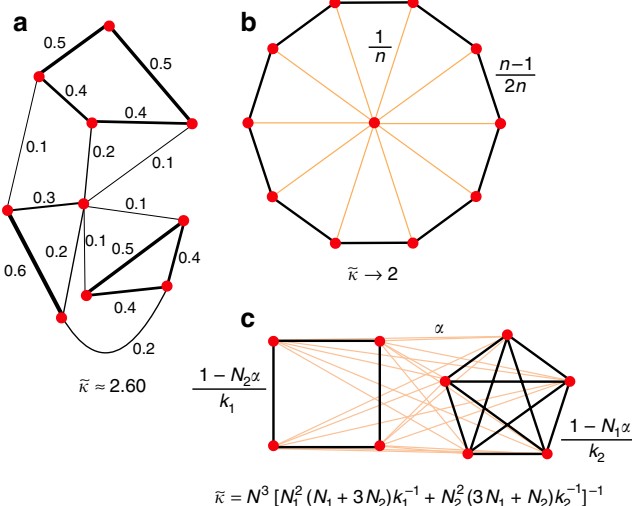

**Fig. 1** Isothermal graphs and their effective degrees. A graph is isothermal if the sum of edge weights is the same for each vertex. The effective degree $\tilde{\kappa}$ of the graph, defined in Eq. (3), determines the outcome of evolutionary game dynamics. **a** An asymmetric isothermal graph; weights are shown for each edge. **b** A wheel graph, with one hub and $n$ wheel vertices. All connections with the hub have weight $1/n$. All connections in the periphery have weight $(n-1)/2n$. As $n \to \infty$, the effective degree approaches 2. A formula for arbitrary $n$ is derived in Supplementary Note 3. **c** A 30-vertex graph generated with preferential attachment[62] and linking number $m = 3$. Isothermal edge weights are obtained by quadratic programming (see Methods). The effective degree, $\tilde{\kappa} \approx 2.47$, is less than the average topological degree, $\bar{k} = 5.6$. **d** An island model, with edges of weight $\alpha \ll 1$ between each inter-island pair of vertices. Shown here are two islands: a $k_1$-regular graph of size $N_1$, and a $k_2$-regular graph of size $N_2$.

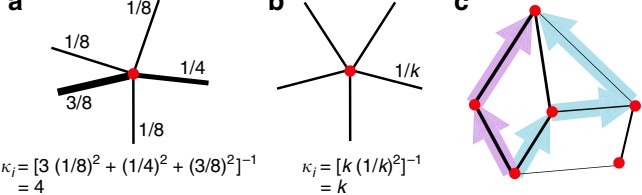

**Fig. 2** Simpson degree and remeeting time. The Simpson degree $\kappa_i = \left( \sum_j w_{ij}^2 \right)^{-1}$ quantifies the effective number (or diversity) of neighbors of a vertex $i$, taking their edge weights into account. **a** If the edge weights to neighbors are nonuniform, the Simpson degree $\kappa_i$ is less than the topological degree $k_i$. Here, $\kappa_i = 4$, which is less than the topological degree, $k_i = 5$. **b** If each neighbor has equal edge weight $1/k$, the Simpson degree is equal to the topological degree, $k$. **c** The remeeting time $\tau_i$ is the expected time for two independent random walks from $i$ to meet each other. The effective degree $\tilde{\kappa}$ of a graph is the weighted harmonic average of the Simpson degrees, with weights given by the remeeting times

degree $\kappa_i$ quantifies the effective number of contacts of individual $i$, accounting for the time spent with each contact, analogously to how the Simpson index of biodiversity[39] quantifies the effective number of species in a population[40]. If all edges from vertex $i$ have equal weight, then the Simpson degree equals the actual number of neighbors: $\kappa_i = k_i$. Otherwise, $\kappa_i$ is less than $k_i$, and decreases as the distribution of edge weights from $i$ becomes more uneven.

Individuals can be one of two types, A or B, corresponding to strategies in the game

$$
\begin{array}{c@{\quad}c}
 & \begin{array}{cc} \text{A} & \text{B} \end{array} \\
\begin{array}{c} \text{A} \\ \text{B} \end{array} & \begin{pmatrix} a & b \\ c & d \end{pmatrix}.
\end{array}
\tag{2}
$$

Each time-step, each individual plays the game with all neighbors. Payoffs from the game are translated into fecundity (reproductive rate) by $F_i = 1 + \delta f_i$, where $f_i$ is the edge-weighted average payoff that $i$ receives from neighbors, and $\delta$ is a parameter quantifying the strength of selection. We study weak selection $(0 < \delta \ll 1)$ as a perturbation of neutral drift $(\delta = 0)$.

Evolution proceeds according to a given update rule. We first consider death-Birth (dB) updating[16]: A vertex $i \in G$ is chosen, with uniform probability, to be replaced. A neighbor $j$ of $i$ is then chosen to reproduce, with probability proportional to $w_{ij}F_j$. The offspring of $j$ replaces the occupant of $i$ and inherits the type of its parent. The capitalization in dB indicates that death is uniform, whereas birth is dependent on payoff. Other update rules are considered later.

Over time, one of the competing types will die out and the other will become fixed. Consider an initial state with a single vertex of type A chosen uniformly at random, and all other vertices of type B. We define the fixation probability $\rho_A$ as the (expected) probability that type A becomes fixed from this initial state. Similarly, $\rho_B$ is the probability that type B becomes fixed from an initial state with one random (uniformly chosen) vertex of type B and all other vertices of type A. We say A is favored if $\rho_A > \rho_B$.

**Condition for success.** We find that the key quantity characterizing an isothermal graph is its effective degree $\tilde{\kappa}$, which we define as a weighted harmonic average of the graph's Simpson degrees:

$$
\tilde{\kappa} = \frac{\sum_i \tau_i}{\sum_i \tau_i \kappa_i^{-1}} = \frac{N^2}{\sum_i \tau_i \kappa_i^{-1}}.
\tag{3}
$$

The weighting $\tau_i$ of vertex $i$ is the expected remeeting time of two random walks that are initialized at $i$ (see Fig. 2c and Methods). Remeeting times arise from tracing ancestries backward in time as coalescing random walks[36,41–43]. If all vertices have $k$ neighbors of equal weight, the effective degree is equal to the topological degree: $\tilde{\kappa} = k$.

We prove in Supplementary Note 1 that strategy A is favored, for death-Birth updating on an isothermal graph under weak selection, if and only if

$$
\sigma a + b > c + \sigma d, \quad \text{with} \quad \sigma = \frac{\tilde{\kappa} + 1 - 4\tilde{\kappa}/N}{\tilde{\kappa} - 1}.
\tag{4}
$$

As an interpretation of Condition (4), consider strategy A to represent cooperation and B to represent defection (noncooperation). For the purposes of this interpretation, we define the cost of cooperation as $C = -\frac{1}{2}(a + b - c - d)$ and the benefit to the partner as $B = \frac{1}{2}(a - b + c - d)$. Then Condition (4) can be algebraically rewritten as

$$
(N/\tilde{\kappa} - 2)B > (N - 2)C.
\tag{5}
$$

The above definitions of benefit $B$ and cost $C$ are motivated by imagining a hypothetical situation in which one's partner is equally likely to be of either type; in this case, playing A rather than B decreases the actor's payoff by $C$ and increases the partner's payoff by $B$. If $B, C > 0$ (cooperation is costly to the actor and beneficial to the recipient) and $\tilde{\kappa} \ll N$, then cooperation is favored as long as $B/C > \tilde{\kappa}$. Well-known results for unweighted $k$-regular graphs[16,17,20,23] are recovered by substituting $k$ for $\tilde{\kappa}$. In contrast, if $\tilde{\kappa} \geq N/2$, then cooperation is never favored, but spiteful behaviors $(B < 0, C > 0)$ can be favored.

According to Conditions (4) and (5), evaluating the conditions for success on a given isothermal graph amounts to computing the effective degree, $\tilde{\kappa}$. This can be done in polynomial time by solving a system of linear equations for coalescence times (see Methods).

**Random isothermal graphs.** How does the effective degree relate to other degree statistics? Since $\tilde{\kappa}$ is a weighted average, it lies between the minimum and maximum Simpson degrees: $\kappa_{\min} \leq \tilde{\kappa} \leq \kappa_{\max}$. However, these bounds are not very informative for strongly heterogeneous graphs.

To gain further insight, we investigated two models for random isothermal graphs. The first, a 2D spatial model (Fig. 3a), is applicable to populations in which each individual occupies a fixed location. An even number of vertices are randomly placed in the unit square. These vertices are repeatedly divided into pairs according to the following scheme: (1) pick an unpaired vertex $i$ uniformly at random; (2) pair $i$ with another unpaired vertex $j$, chosen with probability proportional to $e^{-\beta d(i,j)}$; (3) repeat until all vertices are paired. Here $d(i, j)$ is the Euclidean distance between $i$ and $j$, and $\beta > 0$ tunes the decay of pairing probability with distance. After a specified number of pairing rounds, the edge weight between any two vertices is defined to be the fraction of rounds they were paired. Since each vertex is paired once per round, the resulting weighted graph is undirected and isothermal.

The second model (Fig. 3b) begins with a random graph topology generated by shifted-linear preferential attachment[44,45], and obtains isothermal weightings on the edges via quadratic programming. While the first model is inspired by spatial biological populations, the second is more applicable to social networks.

For both random graph models, we compared $\tilde{\kappa}$ to the (unweighted) arithmetic and harmonic average Simpson degrees (denoted $\kappa_A$ and $\kappa_H$, respectively) as well as to the arithmetic average topological degree $\bar{k}$. We find (Fig. 3c, d) that the harmonic average Simpson degree $\kappa_H$ provides the best estimate in most cases. Therefore, $B/C > \kappa_H$ closely approximates the condition for cooperation to be favored for weak selection on a large isothermal graph. This is significant for the evolution of cooperation, because the harmonic mean of a set of numbers is dominated by its smallest elements. Consequently, the presence of large-degree hubs need not preclude support for cooperation, even if they lead to a large arithmetic average degree (either topological or Simpson).

**Spectral gap bounds for expander graphs.** To formalize the relationship between the effective degree $\tilde{\kappa}$ and the harmonic average Simpson degree $\kappa_H$, we derive bounds on $\tilde{\kappa}$ in terms of the spectral gap—the difference between the two largest eigenvalues of the adjacency matrix. Large graphs with non-negligible spectral gap are called expander graphs, and have important applications in mathematics[38] and computer science[37]. For our purposes, we define a 'family of isothermal expander graphs' as a sequence of isothermal graphs with sizes tending to infinity and

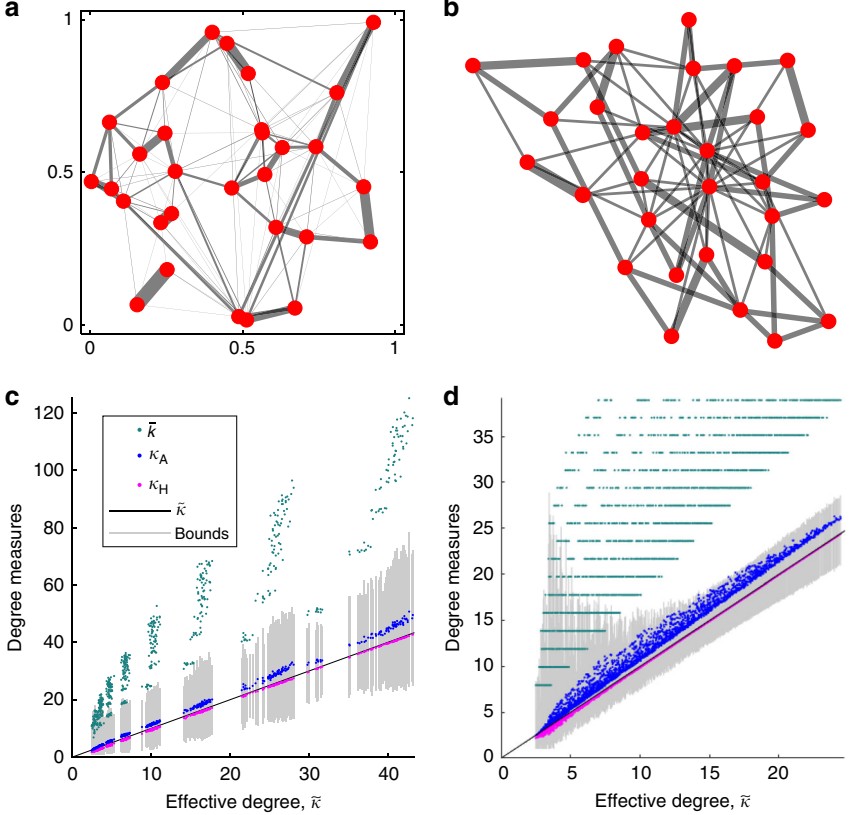

**Fig. 3** Effective degrees of random graphs. **a** A 2D spatial model in which individuals are randomly placed in the unit square and then randomly paired according to their distance from each other. **b** A shifted-linear preferential attachment network[44,45], with isothermal edge weights obtained by minimizing $\sum_{i,j} w_{ij}^2$ subject to $\sum_j w_{ij} = 1$ for all $i$. **c, d** For each graph generated by these two models, the effective degree, $\tilde{\kappa}$ (black line) is plotted against the arithmetic mean topological degree, $\bar{k}$ (green dots); the arithmetic mean Simpson degree, $\kappa_A$ (blue dots); and the harmonic mean Simpson degree, $\kappa_H$ (magenta dots). Vertical gray bars show the quantile bounds (6) for each graph. In almost all cases, $\kappa_H$ provides the best estimate for $\tilde{\kappa}$; $\kappa_H$ performs better only when $\tilde{\kappa}$ is very small. Note that the arithmetic mean topological degree, $\bar{k}$, is significantly larger than the other degree measures in almost all cases. See Methods for further details and parameter values

spectral gap tending to a positive value $g$, which necessarily lies in the range $0 < g \leq 1$ (see Methods).

We prove in Supplementary Note 2 that the remeeting times for such a family are asymptotically bounded by $\tau_i \leq N/g$ for each vertex $i$. We apply this result to bound the effective degree $\tilde{\kappa}$. Let $\kappa_{[a,b]}$ denote the harmonic mean of the Simpson degrees lying between the $a$th and $b$th quantiles, for $0 \leq a < b \leq 1$. For example, $\kappa_{[0,1/4]}$ denotes the harmonic mean over the smallest fourth (first quartile) of Simpson degrees. For a family of isothermal expander graphs, we prove the following asymptotic bounds:

$$\kappa_{[0,g]} \leq \tilde{\kappa} \leq \kappa_{[1-g,1]}. \tag{6}$$

As $g$ increases, both bounds become closer to $\kappa_{[0,1]} = \kappa_H$. Combining (6) with classical inequalities, we obtain the looser but simpler bounds

$$g\kappa_H \leq \tilde{\kappa} \leq \frac{\kappa_A}{g}. \tag{7}$$

**Promoters of cooperation with infinite average degree.** Our results allow us to construct families of isothermal graphs that favor the evolution of cooperation even as the average degree (either topological or Simpson) diverges to infinity.

Let us first consider island-structured populations (Figs. 1d and 4). The islands are represented by isothermal, vertex-transitive graphs, $G_1, \ldots, G_n$, which may differ in their size and network

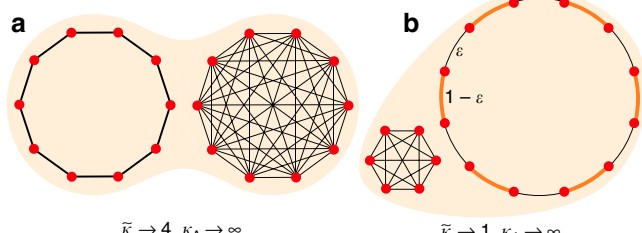

**Fig. 4** Island-structured "super-promoters" of cooperation. We use the island model (Fig. 1c) to construct families of graphs whose effective degree $\tilde{\kappa}$ remains finite while the arithmetic average degree (both Simpson and topological) diverges to infinity. **a** Two islands of equal size: a cycle and a complete graph. As $N \to \infty$, the effective degree $\tilde{\kappa}$ converges to 4, which is the harmonic mean of 2 and infinity. **b** A small complete graph and a large cycle with alternating edge weights, $\varepsilon \ll 1$ and $1 - \varepsilon$. Under the appropriate combination of limits, $\tilde{\kappa}$ converges to 1—meaning that all cooperative behaviors with $B > C > 0$ are favored—while $\kappa_A$ diverges. Calculations are provided in the Methods

structure. An overall isothermal graph $G$ is formed by joining each inter-island pair of vertices by an edge of weight $\alpha \ll 1$, and rescaling intra-island edge weights correspondingly (see Methods). We prove that, if the island sizes are equal, the effective degree $\tilde{\kappa}$ of $G$ is the (unweighted) harmonic mean of the Simpson degrees $\kappa_1, \ldots, \kappa_n$ of the separate islands. If the islands have

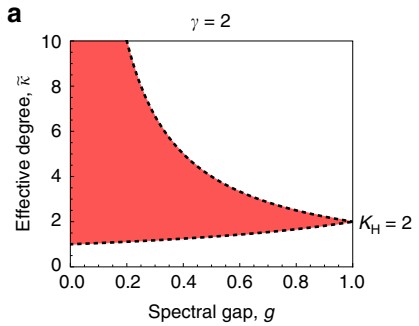
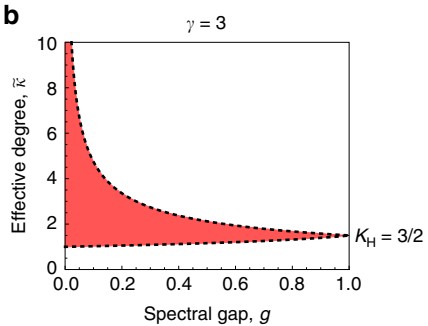

**Fig. 5** Bounds on effective degree for power-law expander graphs. We consider a large isothermal graph for which the Simpson degree distribution is described by the density $f(\kappa) \propto \kappa^{-\gamma}$ on the range $[\kappa_0, \infty)$. The upper and lower bounds (8) are shown for **a** $\gamma = 2$ and **b** $\gamma = 3$, both with $\kappa_0 = 1$. As $g$ approaches 1, the upper and lower bounds both converge to the (unweighted) harmonic mean Simpson degree $\kappa_H$.

different sizes, $\tilde{\kappa}$ is a weighted harmonic mean of $\kappa_1, \ldots, \kappa_n$, with weights depending only on the islands' sizes.

Suppose that one island is a cycle and the other a complete graph of equal size (Fig. 4a). Then as population size increases, the arithmetic mean Simpson degree $\kappa_A$ diverges to infinity, while the effective degree converges to 4. Support for cooperation can be further increased by varying the island sizes and the edge weights of the cycle (Fig. 4b). In the most extreme limit, we have $\kappa_A \to \infty$ but $\tilde{\kappa} \to 1$, meaning that any cooperative behavior with $B > C > 0$ is favored.

Second, we consider a hypothetical family of isothermal expander graphs whose limiting Simpson degree distribution is described by the power-law density $f(\kappa) \propto \kappa^{-\gamma}$, on the range $\kappa_0 \leq \kappa < \infty$, for arbitrary $\gamma \geq 2$ and $\kappa_0 \geq 1$. Evaluating (6) for the corresponding quantile function, $\kappa(x) = \kappa_0(1 - x)^{-1/(\gamma-1)}$, yields (Supplementary Note 3)

$$\left(\frac{\gamma}{\gamma-1}\right)\frac{\kappa_0 g}{1 - (1-g)^{\gamma/(\gamma-1)}} \leq \tilde{\kappa} \leq \left(\frac{\gamma}{\gamma-1}\right)\kappa_0 g^{-1/(\gamma-1)}. \quad (8)$$

These bounds are illustrated in Fig. 5. For $\gamma = 2$, the arithmetic average Simpson degree $\kappa_A$ diverges to infinity, but the upper bound on $\tilde{\kappa}$ is $2\kappa_0/g$. Thus $B/C > 2\kappa_0/g$ is sufficient for cooperation to be favored.

**Other update rules**. So far, we have considered only death-Birth updating. One may also consider Birth-death (Bd) updating[16]: An individual $i$ is chosen, proportionally to its fecundity $F_i$, to reproduce; the offspring of $i$ replaces neighbor $j$ with probability $w_{ij}$. Alternatively, one may let selection act on mortality, leading to two further update rules[20–22]. For Death-birth (Db) updating, an individual $i$ is chosen to die, proportionally to $F_i^{-1}$; a neighbor $j$ is then chosen to reproduce into the vacancy, proportionally to $w_{ij}$. For birth-Death (bD) updating, an individual is chosen to reproduce, with uniform probability; the offspring displaces a neighbor $j$ with probability proportional to $w_{ij}F_j^{-1}$.

We find (Supplementary Note 1) that bD updating leads to the same condition for success as for dB, Eq. (4). In contrast, for Bd or Db, type A is favored for weak selection if and only if

$$\sigma a + b > c + \sigma d, \quad \text{with} \quad \sigma = (N-2)/N. \quad (9)$$

This same condition for success was previously derived for well-mixed populations, under a variety of update rules, with arbitrary selection strength and mutation rate[46–48]. Thus isothermal graph structure has no effect on the conditions for evolutionary game success under weak selection. Rewriting Condition (9) as

$$-(N-1)C - B > 0, \quad (10)$$

we find that cooperation is never favored for positive $B$ and $C$.

This generalizes, to all isothermal graphs, the previous finding that Bd and Db updating do not support cooperation on regular graphs[16,17,20,22,23,26].

The equivalence of success conditions between dB and bD, and between Bd and Db, was previously observed for vertex-transitive graphs[20–22], but does not hold for arbitrary graphs[49]. Here we have demonstrated these equivalences for all isothermal graphs. These equivalences are related to the scales of spatial competition induced by the various update rules. For Bd and Db, a type is favored if it has higher payoff, on average, than its immediate (one-step) neighbors. Having neighbors of high payoff increases the likelihood of being replaced by their offspring (for Bd), or decreases the likelihood of them providing a vacancy to reproduce into (for Db). In contrast, for dB and bD, a type is favored if it higher payoff, on average, than its two-step neighbors. This is because one competes with one's two-step neighbor to fill a vacancy (for dB) or to avoid being replaced (for bD). These observations are made precise in Eq. (14) of the Methods.

**Diffusible public goods**. So far we have assumed that game interactions occur only between immediate neighbors. However, many microbial populations exhibit a form of cooperation in which some cells produce useful chemicals that diffuse through the environment and are utilized by other cells[50,51]. These chemicals may be termed "diffusible public goods"—examples include iron chelators[52,53], hydrolyzed sugars[54], antibiotic resistance agents[55], and growth factors in tumor cells[56]. Public goods production can be exploited by "cheaters", who utilize the good without producing it, leading to a social dilemma[57,58].

We model the diffusion of public goods as a random walk on $G$, starting at the vertex where the good is produced. A benefit $b_n \geq 0$ goes to the vertex at the $n$th step of this walk. That is, benefit $b_0$ goes immediately to the producer, benefit $b_1$ to a random neighbor, benefit $b_2$ to a random neighbor-of-neighbor (which may again be the producer itself), and so on. For the sake of generality, we do not assume any particular form for the $b_n$, only that the total benefit $B = \sum_{n=0}^{\infty} b_n$ is finite. Of the total benefit from public goods produced at vertex $i$, the fraction received by vertex $j$ is $\phi_{ij} = \frac{1}{B}\sum_{n=0}^{\infty} p_{ij}^{(n)} b_n$, where $p_{ij}^{(n)}$ is the probability that a random walk from $i$ visits $j$ at the $n$th step. The cost to produce the good is $C > 0$.

Whether production of diffusible goods is favored depends on the costs to produce, the amount of benefit, the pattern of diffusion, and the spatial structure[57,58]. For Bd or Db updating on an isothermal graph, we show in Supplementary Note 4 that production is favored under weak selection if

$$-C(N-1) + B(N\phi^{(0)} - 1) > 0. \quad (11)$$

Here $\phi^{(0)} = \sum_{i \in G} \frac{\tau_i}{N^2} \phi_{ii}$ is the average amount that a producer benefits from its own good, with the producing vertex weighted by remeeting time. According to this condition, production is favored only if it provides a net benefit to the producers themselves ($B\phi^{(0)} > C$, in a large population).

In contrast, for dB or bD updating, production is favored if

$$-C(N-2) + B[N(\phi^{(0)} + \phi^{(1)}) - 2] > 0. \qquad (12)$$

Above, $\phi^{(1)} = \sum_{i \in G} \frac{\tau_i}{N^2} w_{ij} \phi_{ij}$ is the expected benefit received by a random neighbor of a producer, with the producing vertex again weighted by remeeting time. In this case, production is favored if the average benefit to a neighbor exceeds the net cost to self ($B\phi^{(1)} > C - B\phi^{(0)}$, in a large population).

In short, production is favored for Bd or Db if there is a net benefit to the producer, and for dB or bD if there is a net benefit to the producer plus a randomly chosen neighbor. These results generalize previous findings for vertex-transitive graphs[57]. If we suppose that benefits go only to immediate neighbors ($b_1 = B$ and all other $b_n = 0$), then $\phi^{(0)} = 0$ and $\phi^{(1)} = 1/\tilde{\kappa}$, whereupon Conditions (11) and (12) reduce to our conditions for two-player games, (10) and (5) respectively.

The distinct outcomes for the different update rules can be traced to their scales of spatial competition (one for Bd or Db, two for bD or dB). We show in Supplementary Note 4 that if spatial competition occurs at scale $m$, then benefits accruing at distances $< m$ from the producer contribute to selection for production, while those accruing at distances $\geq m$ are canceled by spatial competition—see Eq. (39) of the Methods.

## Discussion
Analytical results for evolutionary games on graphs have recently been extended from regular graphs to arbitrary weighted graphs[30–32]. Isothermal graphs represent an important class of intermediate generality: flexible enough to represent a wide range of population configurations, yet simple enough to yield powerful results. They represent scenarios in which individuals may differ in their number of connections, but are equal in their reproductive value. Isothermal graphs arise naturally from supposing that all individuals devote equal time to interaction, and can be generated via pairing schemes such as we consider in Fig. 3a.

For dB and bD updating, the condition for success under weak selection takes a simple form, Condition (4), in which the consequences of graph structure are summarized in the effective degree $\tilde{\kappa}$. For large graphs, cooperation is favored if it provides more than a $\tilde{\kappa}$-fold benefit relative to the cost.

Our result allows us to identify graphs (e.g. Figs. 4 and 5a) for which the effective degree remains bounded while the arithmetic average degree (both topological and Simpson) diverges to infinity. This is possible because $\tilde{\kappa}$ is a weighted harmonic average, and harmonic averages (unlike arithmetic averages) are dominated by the smallest elements of a set.

The weights in $\tilde{\kappa}$ are given by the time, $\tau_i$, for two independent random walks from vertex $i$ to rejoin each other. Using spectral graph theory, we derived bounds on these remeeting times, and in turn on the effective degree $\tilde{\kappa}$, in terms of the graph's degree distribution and spectral gap. The appearance of the spectral gap suggests an intriguing link between evolutionary game theory and the theory of expander graphs. Currently, expansion properties are much better understood for regular graphs than for non-regular graphs[36–38]. Isothermal graphs may serve as a useful intermediate class for generalizing expander graph theory.

For Bd or Db updating, the conditions for $\rho_A > \rho_B$ under weak selection are independent of the graph structure. In particular, for these update rules, isothermal graph structure does not promote

the evolution of cooperation, relative to the baseline case of a well-mixed population. These results underscore the principle—previously observed in homogeneous population structures[16,17,21,22,59]—that, for spatial structure to support cooperation, the benefits of cooperation must be distributed at distances less than the scale of competition. Here we have extended this principle to isothermal graphs, with diffusible public goods providing the clearest illustration. Our findings for Bd or Db and are reminiscent of the Isothermal Theorem[6,35], which states that the fixation probability of a mutation of constant fitness, for Bd or Db on any isothermal graph, is the same as in a well-mixed population. The common thread is that, for Bd or Db on isothermal graphs, key aspects of the evolutionary process are invariant with respect to spatial structure. Importantly, for non-isothermal graphs, Condition (9) is not generally valid, and the conditions for success under Birth-death updating vary from graph to graph[60]. It therefore appears that the cancellation principle observed here and in previous work[16,17,21,22] is specific to isothermal graphs. The question of whether Bd or Db updating can promote cooperation on any (non-isothermal) weighted graph remains open.

Our work adds an important nuance to our understanding of the evolution of cooperation. Previous work on regular graphs[16–18,20,22–29] showed that cooperation thrives (for dB or bD updating) when each individual has few neighbors, relative to the overall population size. Condition (4) shows that it is not the raw number of neighbors that matters, but their effective number, as quantified by $\tilde{\kappa}$. Even in highly interconnected societies, cooperation can flourish if most individuals interact primarily with a few close contacts, rather than many loose acquaintances.

## Methods
**Model**. We denote the state of the process by a vector **x**, with entries $x_i$ indicating the type of each vertex $i \in G$: $x_i = 1$ if $i$ has type A and $x_i = 0$ if $i$ has type B. The payoff to vertex $i$ in state **x** is denoted $f_i(\mathbf{x})$, and the fecundity is given by $F_i(\mathbf{x}) = 1 + \delta f_i(\mathbf{x})$.

The four update rules we consider are defined by the probability $e_{ij}(\mathbf{x})$ that the offspring of vertex $i$ replaces the occupant of vertex $j$ in state **x**:

$$e_{ij}(\mathbf{x}) = \begin{cases} \frac{1}{N}\left(\frac{w_{ij}F_i(\mathbf{x})}{\sum_{k \in G} w_{kj}F_k(\mathbf{x})}\right) & \text{death} - \text{Birth (dB)} \\[2ex] \left(\frac{(F_j(\mathbf{x}))^{-1}}{\sum_{k \in G}(F_k(\mathbf{x}))^{-1}}\right)w_{ij} & \text{Death} - \text{birth (Db)} \\[2ex] \left(\frac{F_i(\mathbf{x})}{\sum_{k \in G} F_k(\mathbf{x})}\right)w_{ij} & \text{Birth} - \text{death}(Bd) \\[2ex] \frac{1}{N}\left(\frac{w_{ij}(F_j(\mathbf{x}))^{-1}}{\sum_{k \in G} w_{ik}(F_k(\mathbf{x}))^{-1}}\right) & \text{birth} - \text{Death (bD)}. \end{cases} \qquad (13)$$

**Analysis of weak selection**. Here we summarize the derivation of our main result; Supplementary Note 1 for a full derivation. The key quantity in analyzing selection is the expected change $\Delta(\mathbf{x})$ in the number of A individuals from state **x**. Based on Eq. (13), we calculate this for weak selection:

$$\Delta(\mathbf{x}) = \begin{cases} \frac{\delta}{N}\sum_{i \in G} x_i\left(f_i(\mathbf{x}) - f_i^{(2)}(\mathbf{x})\right) + \mathcal{O}(\delta^2) & \text{dB or bD} \\[2ex] \frac{\delta}{N}\sum_{i \in G} x_i\left(f_i(\mathbf{x}) - f_i^{(1)}(\mathbf{x})\right) + \mathcal{O}(\delta^2) & \text{Bd or Db}. \end{cases} \qquad (14)$$

Above, we have introduced the notation $f_i^{(n)}(\mathbf{x}) = \sum_j p_{ij}^{(n)} f_j(\mathbf{x})$ for the expected payoff of an individual at the terminus of an $n$-step random walk from $i$, where $p_{ij}^{(n)}$ denotes the probability that such a random walk terminates at $j$.

Theorem 4 of Allen and McAvoy[61] implies that $\rho_A > \rho_B$ if and only if $\langle\Delta\rangle > 0$, where $\langle\ \rangle$ denotes an expectation over a particular probability distribution over states, called the "rare-mutation conditional (RMC) distribution". Combining with Eq. (14), type A is favored under weak selection if and only if

$$\sum_{i \in G}\left\langle x_i\left(f_i(\mathbf{x}) - f_i^{(m)}(\mathbf{x})\right)\right\rangle > 0, \qquad (15)$$

where $m = 1$ for Bd or Db, and $m = 2$ for dB or bD.

**Coalescence and remeeting times**. We compute the expectation in Eq. (15) using coalescence times, defined by the recurrence relations

$$\tau_{ij} = \begin{cases} 0 & i = j \\ 1 + \frac{1}{2}\sum_{k \in G}\left(w_{ik}\tau_{jk} + w_{jk}\tau_{ik}\right) & i \neq j. \end{cases} \quad (16)$$

Coalescence times are related to expectations over the RMC distribution by

$$\tau_{ij} \propto \frac{1}{2} - \langle x_i x_j \rangle, \quad (17)$$

for all pairs $i, j \in G$.

The remeeting time $\tau_i$ is the expected time for two independent random walks from $i$ to rejoin each other. It is obtained by the relation

$$\tau_i = 1 + \sum_{j \in G} w_{ij}\tau_{ij}. \quad (18)$$

Remeeting times on isothermal graphs satisfy[30]

$$\sum_{i \in G} \tau_i = N^2. \quad (19)$$

We denote the expected coalescence time from the two ends of an $n$-step random walk as $\tau^{(n)} = \frac{1}{N}\sum_{i,j \in G}p_{ij}^{(n)}\tau_{ij}$. The $\tau^{(n)}$ satisfy the recurrence relation

$$\tau^{(n+1)} = \tau^{(n)} + \frac{1}{N}\sum_{i \in G}p_{ii}^{(n)}\tau_i - 1. \quad (20)$$

We observe that $p_{ii}^{(0)} = 1$, $p_{ii}^{(1)} = 0$ (no self-loops), and $p_{ii}^{(1)} = 1/\kappa_i$ for each $i$. Using Eqs. (3), (19), and (20), we obtain

$$\tau^{(1)} = N - 1 \quad (21)$$

$$\tau^{(2)} = N - 2 \quad (22)$$

$$\tau^{(3)} = N + N/\tilde{\kappa} - 3. \quad (23)$$

**Conditions for success**. We temporarily assume that the game satisfies $a + d = b + c$; we will later show this assumption to be unnecessary. With this assumption, the payoff differences in Eq. (14) can be written as

$$f_i(\mathbf{x}) - f_i^{(n)}(\mathbf{x}) = -C\left(x_i - x_i^{(n)}\right) + B\left(x_i^{(1)} - x_i^{(n+1)}\right), \quad (24)$$

where $x_i^{(n)} = \sum_j p_{ij}x_j$ is the expected type at the end of an $n$-step random walk from $i$, and $B = \frac{1}{2}(a - b + c - d)$ and $C = -\frac{1}{2}(a + b - c - d)$ as in the main text. Eq. (17) implies

$$\sum_{i \in G}\left\langle x_i\left(x_i^{(n)} - x_i^{(m)}\right)\right\rangle \propto \tau^{(m)} - \tau^{(n)}. \quad (25)$$

Applying Eqs. (15), (24), and (25), type A is favored under weak selection if and only if

$$-C\tau^{(2)} + B\left(\tau^{(3)} - \tau^{(1)}\right) > 0, \quad (26)$$

for dB or bD updating. Substituting from Eqs. (21)–(23) yield Conditions (5) of the main text. For Bd or Db, we obtain

$$-C\tau^{(1)} + B\left(\tau^{(2)} - \tau^{(1)}\right) > 0, \quad (27)$$

and substituting from Eqs. (21)–(22) yields the condition $-(N-1)C - B > 0$. Finally, the Structure Coefficient Theorem[60] shows that the assumption $a + d = b + c$ can be dropped, and Conditions (4) and (9) follow.

**Spectral gap and expander graphs**. The spectral gap of an isothermal graph is $g = 1 - \lambda_2$, where $\lambda_2$ is the second-largest eigenvalue of the adjacency matrix. In Supplementary Note 2 we prove the following result:

**Theorem 1.** *On an isothermal graph $G$ of size $N$ and spectral gap $g$, the remeeting time $\tau_i$ for each vertex $i \in G$ is bounded by*

$$\tau_i \leq \frac{N-1}{g} + \frac{2N-1}{N}. \quad (28)$$

We formally define a 'family of isothermal expander graphs' as a sequence of isothermal graphs $\{G_j\}_{j=1}^{\infty}$, with corresponding sizes $N_j$ and spectral gaps $g_j$, such that $N_j \to \infty$ and $\lim_{j \to \infty} g_j = g$, with $0 < g \leq 1$. Then the upper bound (28) is asymptotically $N/g + \mathcal{O}(1/N)$.

Suppose that, as $j \to \infty$, the Simpson degree distribution converges, pointwise in its quantile function, to some continuous function $\kappa(x)$. This means that, in the limiting distribution, a fraction $x$ of Simpson degrees are less than or equal to $\kappa(x)$, for all $0 \leq x \leq 1$. Then the harmonic average Simpson degree over the quantile

range $[a, b]$ is defined as

$$\kappa_{[a,b]} = (b - a)\left(\int_a^b (\kappa(x))^{-1}dx\right)^{-1}. \quad (29)$$

Bounds on the effective degree are obtained by considering all the ways the weights $\tau_i$ can be apportioned among vertices, subject to the constraints of (19) and Bound (28). Placing the maximum weight on the vertices of largest Simpson degree yields $\tilde{\kappa} \leq \kappa_{[1-g,1]}$, while placing the maximum weight on the vertices of smallest Simpson degree yields $\tilde{\kappa} \geq \kappa_{[0,g]}$. Bounds (7) follow from

$$\kappa_{[0,g]} = g\left(\int_0^g (\kappa(x))^{-1}dx\right)^{-1} \leq g\left(\int_0^1 (\kappa(x))^{-1}dx\right)^{-1} = g\kappa_{\mathrm{H}}, \quad (30)$$

and, using the harmonic-arithmetic means inequality,

$$\kappa_{[1-g,1]} \leq \frac{1}{g}\int_{1-g}^1 \kappa(x)\ dx \leq \frac{1}{g}\int_0^1 \kappa(x)\ dx = \frac{\kappa_{\mathrm{A}}}{g}. \quad (31)$$

**Random graph experiments**. For the spatial model (Fig. 3a, c), the population size is $N = 200$, the decay parameter is $\beta = 2^{\ell/2}$ for $\ell = 6, 7, \ldots, 14$, and the number of pairing rounds is $100, 200, \ldots, 800$. Ten random isothermal graphs were generated for each parameter combination. The clusters of points in Fig. 3c correspond to different values of $\beta$.

The shifted-linear preferential attachment model[44,45] (Fig. 3b, d) is defined as follows: Starting from a complete graph of size $m + 1$, new vertices were added one at a time, each linking to $m$ existing vertices, chosen with probability proportional to $k - am$, where $k$ is vertex degree and $a$ is a shift parameter. The process was iterated until the graph reached size $N = 400$. We used linking numbers $m = 4, 5, \ldots, 20$, and shift parameter $a$ varying from 0 to 0.9 in increments of 0.05. For each combination of $a$ and $m$, we generated ten graph topologies. For each topology generated this way, an isothermal weighting was obtained by minimizing $\sum_{i,j}w_{ij}^2$ under the constraint $\sum_j w_{ij} = 1$ for all $i$, using a numerical quadratic programming algorithm. This sum-of-squares minimization was chosen in order produce a unique set of edge weights that are relatively even—and therefore have relatively large Simpson degrees—given the constraints imposed by the topology and isothermality. Graph topologies that could not be made isothermal were removed from the ensemble; such graphs arose for small $m$ and $a$ close to 1 (see Supplementary Note 5 for further discussion). The horizontal bands for $\bar{k}$ in Fig. 3D correspond to particular values of $m$.

**Island model**. The island model is obtained by joining seperate isothermal graphs $G_1, \ldots, G_n$ of respective sizes $N_1, \ldots, N_n$. Each inter-island pair of vertices is joined by an edge of weight $\alpha \ll 1$. Edge weights within each island $G_x$ are then rescaled by $1 - \alpha(N - N_x)$ so that the sum of edge weights at each vertex remains 1. We show in Supplementary Note 3 that, in the limit $\alpha \to 0$, coalescence times within each island $G_x$ are determined by

$$\tau_{ij} = \begin{cases} 0 & i = j \\ 1 + \sum_{y=1}^n N_y T_{xy} + \frac{1}{2}\sum_{k \in G_x}\left(w_{ik}\tau_{jk} + w_{jk}\tau_{ik}\right) & i \neq j. \end{cases} \quad (32)$$

Above, the $T_{xy}$, for $x, y \in \{1, \ldots, n\}$, are themselves the solution to the system

$$T_{xy} = \begin{cases} 0 & x = y \\ \frac{1}{N} + \frac{1}{2N}\sum_{z=1}^n N_z\left(T_{xz} + T_{yz}\right) & x \neq y. \end{cases} \quad (33)$$

Solving Eqs. (32)–(33) yields the coalescence times, from which the remeeting times can be obtained from (18), and the effective degree can be obtained from (3) of the main text. We have obtained a closed-form expression for $\tilde{\kappa}$ in two cases. First, if all islands have equal size, the effective degree is the unweighted harmonic mean of the Simpson degrees on the separate islands: $\tilde{\kappa} = n\left(\sum_{x=1}^n \kappa_x^{-1}\right)^{-1}$. Second, if there are $n = 2$ islands, the effective degree is

$$\tilde{\kappa} = N^3\left(N_1^2(N_1 + 3N_2)\kappa_1^{-1} + N_2^2(3N_1 + N_2)\kappa_2^{-1}\right)^{-1}. \quad (34)$$

For the case of Fig. 4A, we have $N_1 = N_2 = N/2$, $\kappa_1 = 2$, and $\kappa_2 = N/2 - 1$, giving $\tilde{\kappa} = 4(N-2)/(N+2)$. For Fig. 5B, we have $\kappa_1 = N_1 - 1$ and $\kappa_2 = 2(1 - 2\varepsilon + 2\varepsilon^2)$. We set $N_1 = aN_2$, substitute in (34), and take the following sequence of limits: first $\varepsilon \to 0$, then $N_2 \to \infty$, then $a \to 0$. Under this limit sequence, $\tilde{\kappa} \to 1$ while $\kappa_{\mathrm{A}} \to \infty$.

**Diffusible public goods**. For diffusible public goods, Condition (15) still applies, but with the modified payoffs

$$f_i(\mathbf{x}) = -Cx_i + \sum_{n=0}^{\infty} b_n x_i^{(n)}. \quad (35)$$

The expected payoff to an individual $m$ random walk steps from vertex $i$ is

$$f_i^{(m)}(\mathbf{x}) = -Cx_i^{(m)} + \sum_{n=0}^{\infty} b_n x_i^{(n+m)}. \quad (36)$$

Condition (15) therefore becomes

$$-C\sum_{i\in G}\left\langle x_i\left(x_i - x_i^{(m)}\right)\right\rangle + \sum_{n=0}^{\infty} b_n\left\langle x_i\left(x_i^{(n)} - x_i^{(n+m)}\right)\right\rangle > 0, \qquad (37)$$

where, as above, $m = 1$ for Bd or Db, and $m = 2$ for dB or bD. Upon applying Eq. (25), the condition becomes

$$-C\tau^{(m)} + \sum_{n=0}^{\infty} b_n\left(\tau^{(n+m)} - \tau^{(n)}\right) > 0. \qquad (38)$$

Applying (20) and the properties of random walks, we show in Supplementary Note 5 that this condition reduces to

$$-C\left(N\sum_{k=0}^{m-1} p^{(k)} - m\right) + B\left(N\sum_{k=0}^{m-1} \phi^{(k)} - m\right) > 0. \qquad (39)$$

Above, we have defined $p^{(k)} = \sum_{i\in G}\frac{\tau_i}{N^2}p_{ii}^{(k)}$ and $\phi^{(k)} = \sum_{i,j\in G}\frac{\tau_i}{N^2}p_{ij}^{(k)}\phi_{ij}$. Both of these quantities refer to a $k$-step random walk with initial vertex $i$ chosen proportionally to remeeting time; $p^{(k)}$ is the probability that such a walk terminates at its origin, and $\phi^{(k)}$ is the expected fraction of public good produced at the intial vertex that would be received at the terminus. We observe that the benefit term of Eq. (39) includes only benefits accruing at distances less than $m$ from the producer. Substituting the appropriate values of $m$, and noting that $p^{(0)} = 1$ and $p^{(1)} = 0$, we obtain Conditions (11) and (12) of the main text.

**Reporting summary**. Further information on research design is available in the Nature Research Reporting Summary linked to this article.

## Data availability
All data generated or analysed during this study are included in this article and its supplementary information files.

## Code availability
Code that supports the findings of this study is available in Zenodo with the identifier https://doi.org/10.5281/zenodo.3462156

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

## Acknowledgements

B.A. is supported by National Science Foundation award #DMS-1715315. We thank Babak Fotouhi for sharing data on Birth-death updating.

## Author contributions

B.A., G.L., and M.A.N. conceived the project. B.A. and G L. analyzed the model. B. A. and M. A.N. wrote the paper.

## Competing interests

The authors declare no competing interests.
