## [Peer Review File · Nature Communications]

Reviewers' Comments:

Reviewer #1:

Remarks to the Author:

I referee this manuscript against the journal's aims and scope, stated on its web page as follows:

"Nature Communications is [...] dedicated to publishing high-quality research in all areas of the biological, health, physical, chemical and Earth sciences. Papers published by the journal aim to represent important advances of significance to specialists within each field."

I am also instructed to ask "What are the major claims of the paper? Are they novel and will they be of interest to others in the community and the wider field?", "Is the work convincing, and if not, what further evidence would be required to strengthen the conclusions?" and "On a more subjective note, do you feel that the paper will influence thinking in the field?"

Measured against this standard, and against my own perception of Nature Communications as a prestigious journal, I am afraid this manuscript is not suitable in its present form. There is considerable novel and interesting content, and I am conscious of the fact that "significance to specialists with each field" is sufficient, and that no general-interest criterion applies as for many other high-profile journals. I do believe that the paper might have the potential to influence the thinking in the field, but at the same time I am also concerned whether the evidence provided actually supports some of the statements and conclusions in the paper. Further, it is not clear to me if or how exactly the paper, in its present form, provides "important advances of significance to specialists within each field." Based on the paper's content, the relevant fields, among the ones listed above, seem to be biological sciences and perhaps physical sciences.

Potential significance to physical sciences:

I first address the potential significance to the physical sciences; applying a broad interpretation, I include mathematics. While the paper contains substantial novel and interesting material on evolutionary processes on networks, I cannot see how the paper, in its present form, would appeal to readers primarily interested in physical or mathematical sciences to a sufficient degree. The style of writing is primarily aimed at those working in the biological and social sciences, and most mathematical details are in the Supplement. I conclude that the target of this paper is the biology audience more than the maths/physical sciences sector. Hence the paper, as it stands, does not meet the criterion of "important advances of significance to specialists" in the mathematical or physical sciences.

Potential significance to the biological sciences:

I again apply the broadest possible interpretation, to include social phenomena in biological societies. While the paper is nicely written overall, and, I re-iterate, there is considerable novel material in it, again I find that the test of "important advance of significance" is not met. This is mainly because the results in the paper apply to only a restricted set of circumstances, and these are not always appropriately stated. There are also some technical issues. I will detail this below. Further, it is not clear what specific biological scenarios the present study would apply to. Can the authors perhaps provide specific examples of biological or social systems, where this would apply, within the current restrictions of the analysis?

Conclusion:

Do not get me wrong. The results in this paper are very nice, and potentially useful. Overall the paper is well written, but I found it irritating to see that important restrictions of the results are not always

clearly stated, and that some statements in the abstract, conclusions etc cannot really be justified from the paper. This makes it hard for me to be more supportive; I would have found a more modest approach more convincing.

As a consequence, I cannot say that the manuscript meets the standards of the journal in its present form. I do not exclude that it can be made suitable, but a major revision would be required, and one would have to investigate the significance, once the technical issues are addressed and the restrictions of the analysis are made more clear. One would then have to see why -- despite these restrictions -- the results are still significant, and in what way.

I see three ways forward: (a) the authors could tone down the language/be more upfront about the restrictions and then make it more clear what exactly the advances are that are made here, and how they are relevant to specific biological or social systems; (b) the authors could reduce the claim of biological relevance, and focus on the mathematics and the formalism. In both cases a manuscript suitable for Nature Communications may result, but I stress the word "may". Or (c), they could choose to make relatively minor changes, again aiming to highlight the restrictions more clearly, and then submit to a less prestigious journal. In all cases the technical issues below need to be addressed.

Detailed comments:

1. I take issue with the way in which some of the results are stated. Generality is claimed, when in reality many of the findings only apply to very restricted situations. For example, the authors say in the abstract "Cooperative behavior is favored on a large isothermal graph if the benefit-to-cost ratio exceeds the effective degree." or "cooperation is favored on a large graph if and only if it provides a $\tilde{\kappa}$ -fold benefit relative to the cost."

This is one central claimed result of the paper. But looking in more detail, this is based on Eq. (5), which in turn seems to be based on the restricted assumptions laid out just before Eq (5), e.g. "suppose that one's partner is equally likely to be either type". This will not generally be true as the process progresses, and neither does it apply in the typical scenario in which one mutant is placed in an otherwise wildtype population. Also, the result only applies to death-birth updating, and this restriction is not highlighted in sentences such as the above. This is stated later in the paper, but if I read the abstract on its own, the statements there are not fully accurate.

In fairness I should add that the authors are very clear that their results only apply in the weak selection limit.

Minor note: In "cooperation is favored on a large graph if and only if it provides a $\tilde{\kappa}$ -fold benefit relative to the cost." you probably mean "if and only if ... it provides at least a $\tilde{\kappa}$ fold ..."

2. Similarly, at several points throughout the paper the authors make statements such as "all effects of graph structure are captured in the effective degree". That may be true for the quantity the authors are interested in, and in restricted circumstances, but it is not true that "all effects" are captured in this way; for example we do not know how mean fixation times, or their distributions are affected.

3. Again similarly, the authors state in the abstract: "As a surprising example, we report graphs of infinite average degree that are nonetheless highly conducive for promoting cooperation." This statement is based on the results contained in Fig. 4, which in turn is from networks of size 200 nodes as stated in the Supplement. So it is not appropriate to say "infinite average degree" -- the authors have not really tested this. The statement the authors make would only be justified, if they could

present analytical results for the limit $N \rightarrow \infty$, or some sort of numerical extrapolation evidence.

It is also worrying that the authors claim to investigate power law degree distributions with a network of such limited size ($N=200$). With such a small graph it is essentially impossible to probe the degree distribution, as multiple decades of degrees would have to be covered. Have the authors actually looked at the degree distribution of the networks they have generated? Over how many decades does the power law distribution extend in their simulations?

Related to this, I am concerned that the authors simply removed networks from the ensemble that "could not be made isothermal". This is hardly a rigorous procedure -- and any bias introduced in this way cannot be controlled. I accept though that there may be no other way of doing this. Is there any way to assess the effects of this intervention?

4. (minor comment) Wording in 3rd sentence of abstract: "General conditions for evolutionary success". While this sounds very nice, it is also rather unspecific. Success?

5. In Section model: "without loss of generality we scale edge weights such that $\sum_j w_{ij}=1$ ". Is it clear that no generality is lost here? Please explain. What is the justification for minimising $\sum_{ij} w_{ij}^2$ (see Methods)? Does this produce a unique set of w_{ij} for each fixed topology? Does it sample all possible assignments of the w_{ij} in a statistically faithful manner?

6. (minor) "The Simpson degree κ_i quantifies the expected number of contacts of individual i , accounting for the time spent with each contact". Please elaborate/explain.

7. (minor) "how rapidly a ball grows in volume with respect to its radius". Ball? What ball? Please explain, or remove.

8. I did not understand the notation of the crossed out " $<$ " symbol. Do you mean $\lambda_2 < 1$?

9. In Fig. 4, how exactly the quantity on the vertical axis is obtained from the simulations? Is this simply measuring $\tilde{\kappa}$ and then inserting into the given relation for $(B/C)^*$? If this is so, then the figure essentially tests whether $\tilde{\kappa}$ is approximated by κ_H ? If the purpose of the figure is to test this, then why then not just plot $\tilde{\kappa}$ versus κ_H ? If there is an additional message, then please explain.

10. The authors test death-birth processes (with local selection) and birth-death updating (with global selection). But there are also other combinations, death-birth with global selection and birth-death with local selection. For some reason these are left out.

11. "This effect arises because, under birth-death updating, any aid given to a neighbor increases the chances of being replaced by that neighbor. Thus the benefits of cooperator assortment are exactly cancelled by local competition among cooperators."

I understand that this interpretation is taken from previous references. What explicit evidence is there that this mechanism is at work in the present case, other than the observation that birth-death updating does not support cooperation on isothermal graphs as studied here? If I am not mistaken, the logic the authors use is as follows: effect A was observed previously in other scenarios and attributed to mechanism X. Here, we observe something similar to A, and so we also attribute this to X. But no actual evidence is given that mechanism X is at work here.

12. I understand that the authors cannot possibly cite and connect to all existing literature on

evolutionary dynamics on complex networks. But I was surprised to see that no connection is made to the following papers:

Evolutionary Dynamics on Degree-Heterogeneous Graphs

T. Antal, S. Redner, and V. Sood

Phys. Rev. Lett. 96, 188104

Voter models on heterogeneous networks

V. Sood, Tibor Antal, and S. Redner

Phys. Rev. E 77, 041121

There may also be other papers in which effective network sizes, dimensions or degrees were introduced for spreading processes on heterogeneous graphs. It may be worth investigating this a little more.

Reviewer #2:

Remarks to the Author:

In this paper the authors consider, as the title suggests, evolutionary games on isothermal graphs. They build on recent research by the same authors and co-workers on coalescence times, using the idea of the "effective degree" of a graph, to show when the first strategy is favoured for general 2 player, 2 strategy matrix games on a graph (and it is good to have this, rather than just considering a prisoner's dilemma type game, as is often the case - I am sometimes a culprit here!). This leads to the pleasing result that cooperative behaviour is favoured on a large isothermal graph if the benefit to cost ratio is larger than the effective degree of the graph. This is a clear and well-written paper, and I only have a few comments, given below.

- 1) The paper contains a number of interesting results, using important properties such as the spectral gap, and the idea of expander graphs. In particular these include the effective degree, and this is a nice concept. Are the authors defining it here (as seems to be the case, but is not clearly stated) or is it taken from earlier work?
- 2) The authors consider two dynamics, which they call death-birth and birth-death. These two dynamics both have selection at the birth event. This is fine as it goes, except that often researchers use a wider set of dynamics, and in particular there are natural equivalents when selection is on the death event. This is not mentioned, and if the reader did not know better, it would not be obvious that there were such alternatives. The dynamics considered would often be called DBB (or dB) and BDB (or Bd), with the corresponding pair being BDD (bD) and DBD (Db). The authors should make this clear.
- 3) The authors show a number of results in the SI, such as condition (4). The calculations here stop before a clear demonstration of the result. There is no restriction on the SI, so why leave the reader to do the work? Please complete this.
- 4) At the bottom of page 6 the authors give an interpretation of condition (4). I believe condition (4), as mentioned above, and the expression in the interpretation, inequality (5), is analogous to that from the donation game in the SI. However, the situation described in the interpretation generally does not hold (cooperators cluster with other cooperators, defectors with defectors, and how likely an interaction is with one type depends upon the overall frequency); so, why does this explanation work?

5) A very small point. In the SI B and C are defined as twice that in the paper (it makes no difference to the analysis, it just seems a small inconsistency in presentation).

A final note: I have seen various interesting pieces of methodology by these authors and co-authors, and it is often hidden away in the SI. I am generally as interested in the methodology as the headline results. There seems to me to be scope for a review paper on methodology for analysing evolutionary graphs, which can highlight the key ways these are tackled.

Reviewers' comments:

Reviewer #1 (Remarks to the Author):

I referee this manuscript against the journal's aims and scope, stated on its web page as follows:

"Nature Communications is [...] dedicated to publishing high-quality research in all areas of the biological, health, physical, chemical and Earth sciences. Papers published by the journal aim to represent important advances of significance to specialists within each field."

I am also instructed to ask "What are the major claims of the paper? Are they novel and will they be of interest to others in the community and the wider field?", "Is the work convincing, and if not, what further evidence would be required to strengthen the conclusions?" and "On a more subjective note, do you feel that the paper will influence thinking in the field?"

Measured against this standard, and against my own perception of Nature Communications as a prestigious journal, I am afraid this manuscript is not suitable in its present form. There is considerable novel and interesting content, and I am conscious of the fact that "significance to specialists with each field" is sufficient, and that no general-interest criterion applies as for many other high-profile journals. I do believe that the paper might have the potential to influence the thinking in the field, but at the same time I am also concerned whether the evidence provided actually supports some of the statements and conclusions in the paper. Further, it is not clear to me if or how exactly the paper, in its present form, provides "important advances of significance to specialists within each field." Based on the paper's content, the relevant fields, among the ones listed above, seem to be biological sciences and perhaps physical sciences.

We thank the reviewer for their thoughtful feedback. It is clear that significant time and care has been put into this report, and for that we are very grateful. This feedback has pushed us to make three major improvements to the manuscript, which we hope will satisfy all of the reviewer's concerns.

First, to show how isothermal graphs are relevant to biological populations, we have introduced a spatially explicit model for generating random isothermal graphs. It begins with a population randomly distributed on a 2D landscape, and then generates isothermal edge weightings based on randomly pairing individuals according to their distance. This model shows how isothermal graphs arise naturally from considering pairwise interactions in a spatially structured population. This spatial model complements the preferential attachment model, which is more appropriate for social networks. (We intend our model to be applicable to both biological and cultural evolutionary processes.)

Second, we have moved beyond pairwise games by introducing a model of diffusible public goods. This form of cooperation, pervasive in microbial populations, involves a beneficial

chemical that is produced by individuals and diffuses throughout the environment. We obtain weak-selection conditions for production of the public good to be favored. Our results for this model shed new light on the principle that, for cooperation to be favored, the spatial scale of competition must exceed that of cooperation.

Third, we have extended our analysis to two more update rules: Death-birth and birth-Death, both of which have selection acting on mortality rather than reproduction.

We have also run additional numerical analyses and significantly rewritten the main text. We are hopeful that these extensions significantly increase the importance of the manuscript to both the physical and biological sciences.

Potential significance to physical sciences:

I first address the potential significance to the physical sciences; applying a broad interpretation, I include mathematics. While the paper contains substantial novel and interesting material on evolutionary processes on networks, I cannot see how the paper, in its present form, would appeal to readers primarily interested in physical or mathematical sciences to a sufficient degree. The style of writing is primarily aimed at those working in the biological and social sciences, and most mathematical details are in the Supplement. I conclude that the target of this paper is the biology audience more than the maths/physical sciences sector. Hence the paper, as it stands, does not meet the criterion of "important advances of significance to specialists" in the mathematical or physical sciences.

We agree that the writing style is aimed at a more biological audience. However, we hope that at least some of our results will be of interest to researchers the mathematical and physical sciences. In particular, the bounds we derive on remeeting times are of potential mathematical interest in their own right. In order to emphasize this contribution without sacrificing readability for a biological audience, we have highlighted these bounds as a theorem in the Methods section. We also hope the new spatial graph model and the diffusible public goods model may be of some interest to network theorists.

Potential significance to the biological sciences:

I again apply the broadest possible interpretation, to include social phenomena in biological societies. While the paper is nicely written overall, and, I re-iterate, there is considerable novel material in it, again I find that the test of "important advance of significance" is not met. This is mainly because the results in the paper apply to only a restricted set of circumstances, and these are not always appropriately stated.

We agree that our assumptions were not always clearly stated along with the corresponding results, and we have carefully addressed these issues, following the reviewer's suggestions.

In some cases, however, we believe that unfortunate phrasing on our part has left the impression that our results require more assumptions than they actually do. We point out these

situations in our responses below, and have rewritten the text to eliminate any such misunderstandings.

There are also some technical issues. I will detail this below. Further, it is not clear what specific biological scenarios the present study would apply to. Can the authors perhaps provide specific examples of biological or social systems, where this would apply, within the current restrictions of the analysis?

We thank the reviewer for raising this important question. There are really two issues to be addressed here.

First, what kinds of population structures can be described by isothermal graphs? We address this question with our new spatial model for random isothermal graphs, which shows how such graphs can arise naturally in a spatially structured population.

Second, what kinds of biological interactions can be described by our model? We address this question with our new model of diffusible public goods. Diffusible public goods dilemmas have been demonstrated in many microbes (E coli, Pseudomonas, yeast, etc). Our results show how production of such goods can be favored in spatially structured populations, depending on the demographic replacement process (update rule) and the parameters describing the public good.

Conclusion:

Do not get me wrong. The results in this paper are very nice, and potentially useful. Overall the paper is well written, but I found it irritating to see that important restrictions of the results are not always clearly stated, and that some statements in the abstract, conclusions etc cannot really be justified from the paper. This makes it hard for me to be more supportive; I would have found a more modest approach more convincing.

Following the reviewer's suggestions, we have carefully revised the work to make clear all assumptions upon which each of our results depend.

As a consequence, I cannot say that the manuscript meets the standards of the journal in its present form. I do not exclude that it can be made suitable, but a major revision would be required, and one would have to investigate the significance, once the technical issues are addressed and the restrictions of the analysis are made more clear. One would then have to see why -- despite these restrictions -- the results are still significant, and in what way.

I see three ways forward: (a) the authors could tone down the language/be more upfront about the restrictions and then make it more clear what exactly the advances are that are made here, and how they are relevant to specific biological or social systems; (b) the authors could reduce the claim of biological relevance, and focus on the mathematics and the formalism. In both cases a manuscript suitable for Nature Communications may result, but I stress the word "may". Or (c), they could choose to make relatively minor changes, again aiming to highlight the

restrictions more clearly, and then submit to a less prestigious journal. In all cases the technical issues below need to be addressed.

We have followed path (a), by more carefully stating our assumptions and results, while also significantly extending our analysis as described above, in order to make the biological relevance more clear. We hope that, with these changes and extensions, the manuscript can be considered suitable for Nature Communications.

Detailed comments:

1. I take issue with the way in which some of the results are stated. Generality is claimed, when in reality many of the findings only apply to very restricted situations. For example, the authors say in the abstract "Cooperative behavior is favored on a large isothermal graph if the benefit-to-cost ratio exceeds the effective degree." or " cooperation is favored on a large graph if and only if it provides a $\tilde{\kappa}$ -fold benefit relative to the cost."

This is one central claimed result of the paper. But looking in more detail, this is based on Eq. (5), which in turn seems to be based on the restricted assumptions laid out just before Eq (5), e.g. "suppose that one's partner is equally likely to be either type". This will not generally be true as the process progresses, and neither does it apply in the typical scenario in which one mutant is placed in an otherwise wildtype population.

On this particular point, we believe that poor phrasing on our part has led to a misunderstanding. To be clear, we do not assume that one's partner is equally likely to be of either type. As the reviewer points out, this is false in our model. The likelihood of having a partner of the same type is quantified by the coalescence times, which can be calculated for any given graph by solving Eq. (11).

Condition (5) is algebraically equivalent to Condition (4) with $C = -(a+b-c-d)/2$ and $B = (a-b-c+d)/2$. No extra assumptions are needed for this equivalence. Therefore, Condition (5) does not require any extra assumptions beyond those specified in the Model section.

The remark about one's partner being equally likely of either type is there only to motivate the interpretation of B as "benefit" and C as "cost". If one is given only a 2x2 game matrix and no other information, it is reasonable to define benefit and cost with reference to an idealized situation in which the two types are equiprobable. However, we do not assume that this idealized situation actually occurs in our model (it doesn't); moreover, the correctness of the condition in Eq. (5) does not depend on this interpretation.

Clearly, our choice of phrasing was unfortunate and led to unnecessary confusion. We have rephrased to make clear that the remark about equiprobable types is only for the purposes of motivation, and is not an assumption of our model (lines 93-101).

Please see also our response to comment 5 of Reviewer 2, and Section 2.5 of the SI.

Also, the result only applies to death-birth updating, and this restriction is not highlighted in sentences such as the above. This is stated later in the paper, but if I read the abstract on its own, the statements there are not fully accurate.

We thank the reviewer for this suggestion. We now clarify, in the abstract and elsewhere, the specific update rules for which each result holds.

In fairness I should add that the authors are very clear that their results only apply in the weak selection limit.

Minor note: In "cooperation is favored on a large graph if and only if it provides a $\tilde{\kappa}$ -fold benefit relative to the cost." you probably mean "if and only if ... it provides at least a $\tilde{\kappa}$ fold ..."

Thank you for this correction. We have rephrased as, e.g., "the benefit to others exceeds $\tilde{\kappa}$ times the cost" (lines 35-36).

2. Similarly, at several points throughout the paper the authors make statements such as "all effects of graph structure are captured in the effective degree". That may be true for the quantity the authors are interested in, and in restricted circumstances, but it is not true that "all effects" are captured in this way; for example we do not know how mean fixation times, or their distributions are affected.

Thank you for pointing this out. We have clarified this throughout the manuscript.

3. Again similarly, the authors state in the abstract: "As a surprising example, we report graphs of infinite average degree that are nonetheless highly conducive for promoting cooperation." This statement is based on the results contained in Fig. 4, which in turn is from networks of size 200 nodes as stated in the Supplement. So it is not appropriate to say "infinite average degree" -- the authors have not really tested this. The statement the authors make would only be justified, if they could present analytical results for the limit $N \rightarrow \infty$, or some sort of numerical extrapolation evidence.

This appears to be another instance of unclear phrasing on our part, for which we apologize. To clarify, the claim is not based on numerical evidence, but on analytical calculations for three specific families of graphs: (1) the same-size island model depicted in what is now Fig. 4A, (2) the different-size island model depicted in what is now Fig. 4B, and (3) graphs with a $\gamma = 2$ power-law distribution of Simpson degrees (now Fig. 5A). For each family, we prove analytically that (in a particular limit) the arithmetic average Simpson degree diverges to infinity while the effective degree remains bounded. The relevant calculations for the respective

examples are shown in Eq. (8), Eq. (25), and the paragraph following Eq. (25). The second example is particularly striking in that the effective degree converges to 1 simultaneously as the arithmetic average Simpson degree diverges to infinity.

We apologize for not stating the basis for this claim more clearly. To eliminate this confusion, we have introduced a new subsection entitled “Promoters of cooperation with infinite average degree”, containing these three examples.

It is also worrying that the authors claim to investigate power law degree distributions with a network of such limited size (N=200). With such a small graph it is essentially impossible to probe the degree distribution, as multiple decades of degrees would have to be covered. Have the authors actually looked at the degree distribution of the networks they have generated? Over how many decades does the power law distribution extend in their simulations?

We acknowledge that it is unjustified to claim that such small networks have power-law degree distribution, and we apologize for this inaccuracy. Since our investigation of these graphs does not depend on them being power-law distributed, we have simply removed any mention of “power law” from the discussion of the shifted-linear preferential attachment networks. The ensemble resulting from this model suffices to probe the relationships between various degree measures (Fig. 3BD). (The analytical calculations for Eq. (8) and Figure 5 are separate from the preferential attachment model, and we have reorganized the text to make this more clear.)

Related to this, I am concerned that the authors simply removed networks from the ensemble that “could not be made isothermal”. This is hardly a rigorous procedure -- and any bias introduced in this way cannot be controlled. I accept though that there may be no other way of doing this. Is there any way to assess the effects of this intervention?

We thank the reviewer for raising this question. Analyzing our numerical data, we find that topologies that were removed are those for which the linking number m is small, and the shift parameter a is close to 1. We have illustrated this in the new Figure S2 of the SI, and have added discussion of this in Section 9 of the SI.

4. (minor comment) Wording in 3rd sentence of abstract: “General conditions for evolutionary success”. While this sounds very nice, it is also rather unspecific. Success?

We have rephrased as “general conditions were derived for a trait to be favored under weak selection”.

5. In Section model: “without loss of generality we scale edge weights such that $\sum_j w_{\{ij\}}=1$ ”. Is it clear that no generality is lost here? Please explain.

No generality is lost because, for each update rule, replacement probabilities depend only on relative (not absolute) edge weights. For dB updating, the probability that a vacant vertex j is

replaced by the offspring of i is $w_{ij} F_i / (\sum_k w_{kj} F_k)$; this probability is unaffected if all edge weights are scaled by a constant. For B_d , if vertex i reproduces, the probability that vertex j dies is $w_{ij} / (\sum_k w_{ik})$; again this probability is invariant with respect to rescaling of edge weights. Similar observations apply to D_b and bD . We have added a brief explanation of this to the SI--see parenthetical remark after Eq. (2).

What is the justification for minimising $\sum_{ij} w_{ij}^2$ (see Methods)? Does this produce a unique set of w_{ij} for each fixed topology? Does it sample all possible assignments of the w_{ij} in a statistically faithful manner?

Minimizing the sum of squared weights produces a set of weights that are relatively even, given the constraints imposed by the topology and isothermality. This was done to produce larger and more varied Simpson degrees, in order to better probe the relationship between the various degree measures. The w_{ij} produced this way are unique (so long as an isothermal weighting exists) since the quadratic cost function is strictly convex. We have added mention of this in the Methods (subsection "Random graph experiments").

6. (minor) "The Simpson degree κ_i quantifies the expected number of contacts of individual i , accounting for the time spent with each contact". Please elaborate/explain.

Here we are drawing an analogy to how the Simpson index of biodiversity quantifies the effective number of species in a population--a concept popularized by conservation biologist Lou Jost. We have rephrased the text to make this clear (lines 56-59).

7. (minor) "how rapidly a ball grows in volume with respect to its radius". Ball? What ball? Please explain, or remove.

We agree this was potentially confusing. Since a precise definition would take too much space and is not necessary for our purposes, we have removed this phrase.

8. I did not understand the notation of the crossed out " \ll " symbol. Do you mean $\lambda_2 \ll 1$?

We had meant that g is not negligible in comparison to 1. Since this notation was evidently ambiguous, we have removed it. We now instead spell everything out in terms of a family of graphs with spectral gap converging to a positive value (lines 143-148).

9. In Fig. 4, how exactly the quantity on the vertical axis is obtained from the simulations? Is this simply measuring $\tilde{\kappa}$ and then inserting into the given relation for $(B/C)^{g^}$? If this is so, then the figure essentially tests whether $\tilde{\kappa}$ is approximated by κ_H ? If the purpose of the figure is to test this, then why then not just plot $\tilde{\kappa}$ versus κ_H ? If there is an additional message, then please explain.*

We thank the reviewer for this suggestion. We have re-run the numerical analysis and redone this figure (now Figure 3) so that the effective degree is plotted against κ_H , along with other degree measures and the spectral gap bounds.

10. The authors test death-birth processes (with local selection) and birth-death updating (with global selection). But there are also other combinations, death-birth with global selection and birth-death with local selection. For some reason these are left out.

Thank you for this suggestion. We have now analyzed these two other combinations, which have elsewhere been termed “Death-birth” and “birth-Death”. These update rules have selection on mortality, in contrast to death-Birth and Birth-death, which have selection on reproduction. (We believe the first usage of this capitalization scheme is Hindersin and Traulsen, 2015).

We show in a new subsection (“Other update rules”) that the conditions for birth-Death are the same as those for death-Birth, and the conditions for Death-birth are the same as those for Birth-death. This had previously been noted for regular graphs; we have now generalized this observation to isothermal graphs.

11. "This effect arises because, under birth-death updating, any aid given to a neighbor increases the chances of being replaced by that neighbor. Thus the benefits of cooperator assortment are exactly cancelled by local competition among cooperators."

I understand that this interpretation is taken from previous references. What explicit evidence is there that this mechanism is at work in the present case, other than the observation that birth-death updating does not support cooperation on isothermal graphs as studied here? If I am not mistaken, the logic the authors use is as follows: effect A was observed previously in other scenarios and attributed to mechanism X. Here, we observe something similar to A, and so we also attribute this to X. But no actual evidence is given that mechanism X is at work here.

We thank the reviewer for encouraging us to think more carefully about this issue. We believe that the issues regarding the scales of competition and cooperation are most clearly seen in the context of diffusible public goods. In this context, we demonstrate that, if competition occurs at scale m , then all benefits accruing at scales m or higher are canceled by spatial competition. For Bd or Db updating, competition occurs at scale 1, so all aid given to neighbors is cancelled. For dB or bD, competition occurs at scale 2, and so cooperation with one-step neighbors can be favored if the benefit/cost ratio is sufficiently large. The results for two-player games can be recovered from the diffusible public goods results by supposing that benefits only go to one-step neighbors.

These issues are now discussed in the last paragraph of “Other update rules”, in the last paragraph of “Diffusible public goods”, after Eq. (39) of the Methods, and in Section 8 of the SI.

12. I understand that the authors cannot possibly cite and connect to all existing literature on evolutionary dynamics on complex networks. But I was surprised to see that no connection is made to the following papers:

Evolutionary Dynamics on Degree-Heterogeneous Graphs

T. Antal, S. Redner, and V. Sood

Phys. Rev. Lett. 96, 188104

Voter models on heterogeneous networks

V. Sood, Tibor Antal, and S. Redner

Phys. Rev. E 77, 041121

There may also be other papers in which effective network sizes, dimensions or degrees were introduced for spreading processes on heterogeneous graphs. It may be worth investigating this a little more.

We apologize for this oversight. We have now cited these works.

Reviewer #2 (Remarks to the Author):

In this paper the authors consider, as the title suggests, evolutionary games on isothermal graphs. They build on recent research by the same authors and co-workers on coalescence times, using the idea of the "effective degree" of a graph, to show when the first strategy is favoured for general 2 player, 2 strategy matrix games on a graph (and it is good to have this, rather than just considering a prisoner's dilemma type game, as is often the case - I am sometimes a culprit here!). This leads to the pleasing result that cooperative behaviour is favoured on a large isothermal graph if the benefit to cost ratio is larger than the effective degree of the graph. This is a clear and well-written paper, and I only have a few comments, given below.

1) The paper contains a number of interesting results, using important properties such as the spectral gap, and the idea of expander graphs. In particular these include the effective degree, and this is a nice concept. Are the authors defining it here (as seems to be the case, but is not clearly stated) or is it taken from earlier work?

We thank the reviewer for highlighting this lack of clarity. We now make clear in the introduction that our notion of effective degree is new.

2) The authors consider two dynamics, which they call death-birth and birth-death. These two dynamics both have selection at the birth event. This is fine as it goes, except that often researchers use a wider set of dynamics, and in particular there are natural equivalents when

selection is on the death event. This is not mentioned, and if the reader did not know better, it would not be obvious that there were such alternatives. The dynamics considered would often be called DBB (or dB) and BDB (or Bd), with the corresponding pair being BDD (bD) and DBD (Db). The authors should make this clear.

We have now extended the analysis to all four update rules: dB, Bd, Db, and bD; see the subsection “Other update rules”.

3) The authors show a number of results in the SI, such as condition (4). The calculations here stop before a clear demonstration of the result. There is no restriction on the SI, so why leave the reader to do the work? Please complete this.

We thank the reviewer for this suggestion, and have completed the explicit derivation in Section 2.5 of the SI.

4) At the bottom of page 6 the authors give an interpretation of condition (4). I believe condition (4), as mentioned above, and the expression in the interpretation, inequality (5), is analogous to that from the donation game in the SI. However, the situation described in the interpretation generally does not hold (cooperators cluster with other cooperators, defectors with defectors, and how likely an interaction is with one type depends upon the overall frequency); so, why does this explanation work?

The reason this works is the Structure Coefficient Theorem of Tarnita et al (2009). In Eq. (27) of the SI, we show that for any evolutionary game process satisfying the conditions of this theorem (which are very general), the condition for success can be written

$$-(\sigma + 1)C + (\sigma - 1)B > 0,$$

with B and C as defined in our manuscript. In fact, this is just an alternate formulation of the Structure Coefficient Theorem. Importantly, B and C depend only on the game matrix, not on the evolutionary process. The actual degree of assortment stemming from the evolutionary process, and the corresponding consequences for selection, are reflected in the value of σ . We now clarify this in the Methods (subsection “Conditions for success”), as well as in Section 2.5 of the SI.

See also our response to comment 1 of Reviewer 1.

5) A very small point. In the SI B and C are defined as twice that in the paper (it makes no difference to the analysis, it just seems a small inconsistency in presentation).

Thank you for this correction.

A final note: I have seen various interesting pieces of methodology by these authors and co-authors, and it is often hidden away in the SI. I am generally as interested in the methodology as the headline results. There seems to me to be scope for a review paper on methodology for analysing evolutionary graphs, which can highlight the key ways these are tackled.

Thank you for this suggestion--this methodology will be included in a book manuscript we are writing!

Reviewers' Comments:

Reviewer #1:

Remarks to the Author:

The authors have addressed the technical queries, and I would like to thank them for their extensive and comprehensive reply to my earlier comments. The main result of this paper is appealing to theorists. The paper is now in a much better shape, and limitations are more clearly stated, the paper is well written. The authors have also added considerable new material.

The authors state in their reply that their (main) target audience are readers in the biological sciences. It is clear that there is also interest to the mathematical sciences.

Given that the biology audience is the main target, one must concede that the paper is relatively theoretical, and in particular that it does not address a specific biological exemplar or case study (the work presented at the end is still fairly theoretical and removed from direct application). At this stage, this is my main reservation and as a consequence it is not obvious (to me) that the main findings of this paper are striking enough to justify publication in Nature Communications. I find it hard to come to a 'yes or no' type of decision. To resolve this, I would need to know how much applicability this journal requires for theoretical work of this general type.

Given that 'general interest' criterion does not apply, and that the required standard is "important advances of significance to specialists" I am probably on the side of 'yes, publish this', interpreting specialists as those working on mathematical models of evolution. I am also conscious that I am asked the question 'On a more subjective note, do you feel that the paper will influence thinking in the field?'. I think it might.

Reviewer #2:

Remarks to the Author:

The authors have put in a lot of work to improve this paper following my comments, and in particular the comments of the other reviewer. I note, concerning one point raised by Reviewer 1, that as a mathematical rather than a biological specialist, their methodologies are certainly interesting to me. In particular the authors have responded to the points I raised well, and I am happy for the paper to be published.

I have one question that I hope they could add a little explanation for, as well as a couple of small points.

In Figure 3, the plot seems to divide into two distinct parts, a cluster of points around the $y=x$ line and a completely different set, with considerable blank space between them. What is happening here and why?

There are a couple of places where the lines of text run on after where they should stop (page 2 line 5 and page 8 line 144).
page 6 line 86 "the the" appears.

REVIEWERS' COMMENTS:

Reviewer #1 (Remarks to the Author):

The authors have addressed the technical queries, and I would like to thank them for their extensive and comprehensive reply to my earlier comments. The main result of this paper is appealing to theorists. The paper is now in a much better shape, and limitations are more clearly stated, the paper is well written. The authors have also added considerable new material.

The authors state in their reply that their (main) target audience are readers in the biological sciences. It is clear that there is also interest to the mathematical sciences.

Given that the biology audience is the main target, one must concede that the paper is relatively theoretical, and in particular that it does not address a specific biological exemplar or case study (the work presented at the end is still fairly theoretical and removed from direct application). At this stage, this is my main reservation and as a consequence it is not obvious (to me) that the main findings of this paper are striking enough to justify publication in Nature Communications. I find it hard to come to a 'yes or no' type of decision. To resolve this, I would need to know how much applicability this journal requires for theoretical work of this general type.

Given that 'general interest' criterion does not apply, and that the required standard is "important advances of significance to specialists" I am probably on the side of 'yes, publish this', interpreting specialists as those working on mathematical models of evolution. I am also conscious that I am asked the question 'On a more subjective note, do you feel that the paper will influence thinking in the field?'. I think it might.

We thank the reviewer for their thoughtful and careful assessment of our work.

Reviewer #2 (Remarks to the Author):

The authors have put in a lot of work to improve this paper following my comments, and in particular the comments of the other reviewer. I note, concerning one point raised by Reviewer 1, that as a mathematical rather than a biological specialist, their methodologies are certainly interesting to me. In particular the authors have responded to the points I raised well, and I am happy for the paper to be published.

I have one question that I hope they could add a little explanation for, as well as a couple of small points.

In Figure 3, the plot seems to divide into two distinct parts, a cluster of points around the $y=x$ line and a completely different set, with considerable blank space between them. What is happening here and why?

The upper cluster of points shows the values of the arithmetic mean topological degree, \bar{k} for each graph. Since \bar{k} is much larger than any of the other degree measures, the values of \bar{k} appear separated from the other measures. We have added a brief note to the figure caption to clarify this.

*There are a couple of places where the lines of text run on after where they should stop (page 2 line 5 and page 8 line 144).
page 6 line 86 "the the" appears.*

We thank you for these corrections. We have corrected the "the the". The overhang is a LaTeX issue that will be presumably be corrected when proofs are generated.